# *Thraustochytrium* sp. and *Aurantiochytrium* sp.: Sustainable Alternatives for Squalene Production

**DOI:** 10.3390/md23030132

**Published:** 2025-03-19

**Authors:** Júnior Mendes Furlan, Graciela Salete Centenaro, Mariane Bittencourt Fagundes, Carlos Borges Filho, Irineu Batista, Narcisa Bandarra

**Affiliations:** 1Chromatography and Food Analysis Research Group, Federal University of Pampa, Itaqui 97650-000, RS, Brazil; gracielacentenaro@unipampa.edu.br (G.S.C.); carlosborgesfilho1991@gmail.com (C.B.F.); 2Interdisciplinary Centre of Marine and Environmental Research CIIMAR, 4450-208 Matosinhos, Porto, Portugal; 3Portuguese Institute for Sea and Atmosphere, 1495-006 Lisbon, Portugal; irineu@ipma.pt (I.B.); narcisa@ipma.pt (N.B.)

**Keywords:** culture system, nitrogen source, squalene, thraustochytrids, sensitivity analysis

## Abstract

This study investigated a sustainable alternative to squalene production utilizing *Thraustochytrium* sp. and *Aurantiochytrium* sp., thereby reducing dependence on critically endangered sharks exploited for this compound. By optimizing fed-batch cultivation, a technique prevalent in industrial biotechnology, we have enhanced squalene yields and have demonstrated, through sensitivity analysis, the significance of this shift in preserving species at risk of extinction. Optimization of culture conditions led to the highest biomass concentrations for *Thraustochytrium* sp. being achieved at lower C–N ratios (<5.0), while the optimal biomass production for *Aurantiochytrium* sp. occurred in culture media with a high C–N ratio of 54:50. Regarding squalene production, *Thraustochytrium* sp. produced 26.13 mg/L in the fed-batch system after 72 h, and *Aurantiochytrium* sp. produced 54.97 mg/L in a batch system with 30 g/L glucose and 0.22 g/L nitrogen after 96 h, showcasing their potential for industrial applications. Moreover, the sensitivity analysis revealed that, on an industrial scale, both strains could produce up to 59.50 t of squalene annually in large-scale facilities, presenting a valuable and sustainable alternative for the biotechnological industry and significantly reducing the reliance on non-renewable and endangered sources such as shark liver oil and preventing the annual capture of over 156,661 sharks.

## 1. Introduction

Squalene (6E,10E,14E,18E,22E)-2,6,10,15,19,23-Hexamethyltetracosa-2,6,10,14,18,22-hexaene (C_30_H_50_), a linear triterpene, holds significant commercial demand for its use as an emollient in pharmaceuticals and cosmetics, its natural antioxidant and antifungal properties, and its capability to inhibit chemically induced tumorigenesis [1,2,3,4]. Despite being synthesized by humans, the levels of squalene decline with age, emphasizing the need for external supplementation. When combined with other bioactive substances, this compound can also act as a key component in nutraceuticals and functional foods that target oxidative stress and age-related diseases [2]. Traditionally, squalene has been extracted from deep-sea shark liver oil, which boasts the highest yields of the compound. However, this practice has led to the overexploitation of shark populations, with an estimated 100 million sharks harvested annually [5,6,7]. It is reported that 90% of the squalene used in cosmetics is derived from shark liver oil [7], posing a serious threat to marine biodiversity and underscoring the urgent need for alternative, sustainable squalene sources.

Heterotrophic protists from the Thraustochytriaceae family have emerged as a promising biotechnological alternative for squalene production. Species such as *Thraustochytrium* sp., *Aurantiochytrium* sp., and *Schizochytrium* sp., known for their exceptional oil production rich in bioactive compounds such as docosahexaenoic acid, carotenoids, and squalene [1,8,9], have shown promise. Among these, *Aurantiochytrium* sp. has achieved the highest squalene yield, with some studies reporting up to 198 mg/g of biomass [10]. *Thraustochytrium* sp., in comparison, has demonstrated high growth rates and is suitable for commercial-scale fermentation, offering a significant potential for toxin-free squalene production [8,11].

The production of squalene in Thraustochytriaceae is highly dependent on culture conditions. Factors such as carbon and nitrogen availability, temperature, pH, salinity, and agitation speed play crucial roles in influencing yield. Previous research has indicated that optimizing these conditions—such as the carbon-to-nitrogen ratio or employing alternative nutrient sources like food waste—can enhance production efficiency [5,12,13,14]. While fed-batch systems have shown potential to improve production, most studies have concentrated on fatty acid profiles rather than on terpenoids like squalene [15,16], underscoring the need for cultivation condition optimization specific to squalene production in Thraustochytriaceae. Given this context, this study aimed to evaluate the impact of various carbon and nitrogen concentrations, as well as cultivation systems (batch vs. fed-batch), on squalene production by *Thraustochytrium* sp. ATCC 26185 and *Aurantiochytrium* sp. PRA-276.

Presenting a novel approach, this study evaluates fed-batch cultivation modes, prevalent in industrial bioprocesses, for squalene production using thraustochytrid strains. By optimizing these methods, our research sets the foundation for future industrial-scale squalene production, offering a sustainable alternative to traditional sources. Additionally, this study assesses regions heavily involved in shark exploitation for squalene extraction, focusing on species such as the Portuguese dogfish (*Centroscymnus coelolepis*) and the Leafscale gulper shark (*Centrophorus squamosus*), both classified as critically endangered [17]. Our findings suggest that adopting microbial production systems could significantly reduce the harvesting of these threatened species, contributing to their conservation. This research aims to develop a sustainable and economically viable production process for squalene, thereby decreasing dependence on environmentally detrimental methods and addressing the global demand for this bioactive compound.

## 2. Results

### 2.1. Influence of Cultivation System on Biomass Productivity

*Thraustochytrium* sp. and *Aurantiochytrium* sp. belong to the same phylum but exhibit different profiles and growth conditions. Results from Thraustochytrid performance are presented in Figure 1.

Regarding the cell culture of *Thraustochytrium* sp., the highest final biomass of 13.9 g/L was achieved under the control conditions (T1), followed by T5, T2, T4, and T3 at 96 h of cultivation (Figure 1a); this strain exhibited a short lag phase. An increase in the initial glucose concentration to T2 negatively impacted the total biomass concentration, resulting in a reduction of over 50%. Furthermore, the reduction of total nitrogen content from 2.4 to 0.8 g/L or its fed-batch supply at a constant rate of 0.009 g/(L·h) also caused a substantial decline in the biomass produced. This production was reduced to less than one-tenth (T3) and over half (T4) of the output achieved in the control treatment (T1). Treatments T2, T3, and T4, which utilized the highest C–N ratio, resulted in the lowest observed biomass yields.

In contrast, *Aurantiochytrium* sp. demonstrated increased cell division and growth under high C–N ratios, leading to higher amounts of final biomass. As shown in Figure 1b, *Aurantiochytrium* sp. ATCC PRA 276 grew rapidly, achieving the highest biomass concentration (23.9 g/L) at 96 h of cultivation. The biomass concentration of *Aurantiochytrium* sp. during batch experiments was found to increase significantly when the initial nitrogen amount was low, as depicted in Figure 1b. The biomass concentration increased by 125% from the control (trial A1) to trial A2 and by 262.1% in trial A3. The continuous supply of glucose (0.14 g/[L·h]) and total nitrogen (0.0014 g/[L·h]) through fed-batch in experiment A4 negatively affected the biomass concentration, which decreased to 13 g/L compared to trial A3 (batch culture system). Despite the continuous replenishment of nitrogen in the medium, the extremely low nitrogen levels were likely too stressful for the biosynthesis of nitrogen-containing compounds, such as proteins and nucleic acids, which are essential for cell division [18]. However, even under these conditions, the final biomass concentration was higher than that in the control treatment.

### 2.2. Influence of Cultivation System on Squalene Productivity

Figure 2 presents the squalene yield (mg/g) and concentration (mg/L) obtained from *Thraustochytrium* sp. cultivated in a bioreactor. Generally, squalene content in the dry biomass was lower in the batch culture system than in the fed-batch system. The highest squalene yields were achieved in trials T5 (3.23 mg/g) and T4 (1.73 mg/g) by using lower glucose levels or their continuous supply or by employing higher total nitrogen levels or their continuous supply.

The maximum squalene concentration observed was 26.13 mg/L after 72 h of cultivation. However, the highest yield of squalene per biomass unit, 3.23 mg/g, was achieved at 48 h (T5). This discrepancy can be attributed to the relationship between squalene concentration (mg/L) and cell concentration in the culture medium. At 48 h, the biomass concentration was 3.5 g/L, whereas at 72 h, it significantly increased to 9.9 g/L. This resulted in a higher total squalene concentration but a lower yield per gram of biomass. Regarding the squalene yield (mg/g) and concentration (mg/L) of *Aurantiochytrium* sp., this strain exhibits distinct kinetic characteristics in the batch system, as illustrated in Figure 3. An increase in squalene content within the cell biomass was observed as the nitrogen concentration in the culture medium decreased. Specifically, squalene content increased from 0.58 mg/g in A1 (3 g/L of total nitrogen) to 2.52 mg/g in A3 (0.22 g/L of total nitrogen). Similarly, A2, with 0.4 g/L of total nitrogen, exhibited an intermediate squalene content of 1.93 mg/g.

Additionally, after 48 h of cultivation, a decrease in squalene content was observed. Nevertheless, the highest concentration of squalene (54.97 mg/L) was recorded after 96 h in treatment A3 as a result of increased biomass production (23.9 g/L). The fed-batch system trial (A4) showed the lowest squalene yield, likely due to a higher consumption rate of glucose and total nitrogen (0.15 g/L·h and 0.0019 g/L·h, respectively) compared to their supply rate (0.14 g/L·h and 0.0014 g/L·h, respectively).

The effects of the tested culture conditions on squalene production indicated that productivity was influenced by several factors. For *Aurantiochytrium*, the most significant factor was the low nitrogen concentration in the batch system. However, for *Thraustochytrium*, the highest squalene content was observed under high nitrogen conditions.

The differences between the strains are evident in their squalene production patterns, both in specific content (mg/g dry biomass) and absolute concentration (mg/L). Strain A3 exhibited the highest squalene accumulation, maintaining elevated levels up to 96 h, which indicates a strong lipid biosynthesis potential under the given conditions. In contrast, strain T5 reached a high peak at 72 h but experienced a sharp decline at 96 h, possibly due to metabolic limitations or degradation processes. The other strains (T1, T2, T3, A1, and A4) produced significantly lower squalene levels, suggesting that they may prioritize alternative metabolic pathways or require different conditions for optimal lipid accumulation. These differences suggest that A3 has a more stable and sustained squalene biosynthesis capability, while T5 exhibits a transient but high production peak. These variations underscore the importance of strain selection in optimizing squalene production, as different strains respond uniquely to environmental and nutritional conditions.

### 2.3. Squalene Sensitivity Analysis

Deep-sea shark liver oil is the most common source of squalene for the pharmaceutical industry [6]. However, due to overfishing, which has led to a decline in the shark population [19], alternative sources must be identified. The shark migration region and fishing activities are presented in Figure 4. Currently, there is control and monitoring of sharks in this region (Figure 4A), a measure necessitated by overfishing, as reported by Kyne et al. [20]. Furthermore, the region around Australia has experienced significant fishing activity over the past five years, according to the GPS tool from the Global Fishing Watch (Figure 4B), despite the implementation of laws and conservation measures.

Sustaining the market for shark oil-based squalene not only poses a threat to marine biodiversity but also represents an economic oversight globally. Climate change has significantly affected sea temperatures, leading to changes in these animals’ migratory patterns, reproduction, and other behavioral aspects, making this market susceptible to economic fluctuations [21]. To mitigate these impending impacts, oleaginous strains present a promising alternative for economic and sustainable production that should be implemented as soon as possible.

The estimated production of squalene in kg/m^3^ per year within algae-based biotechnological industries of varying sizes and using different biotechnological treatments is illustrated in Figure 5 and Figure 6, which pertain to *Thraustochytrium* sp. and *Aurantiochytrium* sp., respectively. We have compared these estimates to the number of sharks required to obtain an equivalent amount of squalene. The shark species included in this comparison were selected based on their extinction risk, as reported by the IUCN Red List Global Assessment (IUCN Red List of Threatened Species). Additionally, various fishing parameters were considered in selecting these sharks, specifically deep-sea species from Australia and New Zealand (Oceania), as these countries are primary exporters of this raw material. Species classified as deep-sea dogfish were chosen due to their specific use for squalene extraction [22]. The liver dimensions of each species, the proportion of oil contained within the livers, and the amount of squalene present in the oil were sourced from the FishBase platform (version 02/2022, Stockholm, Sweden) [23] and the literature [24,25,26].

The biotechnology industry has been expanding its potential, and many currently commercialized bioactive compounds are derived from various strains. The use of Thraustochytrids strains for squalene production may offer a sustainable alternative to reducing shark fishing. Considering an industrial capacity of 1000 m^3^ and using the *Thraustochytrium* sp. strain, which demonstrated the highest squalene productivity measured in kg/m^3^/year (T5), as shown in Figure 5, it is evident that fishing 8366.18 Portuguese dogfish (Centroscymnus coelolepis) sharks would be necessary. This species is known to have a lower amount of liver oil and squalene compared to the Birdbeak dogfish (*Deania calceus*), which would require a catch of 4567.74 sharks. The numbers grow significantly when considering an industry with a capacity exceeding 10,000 m^3^, reaching values of 83,661.84 sharks.

Regarding *Auratiochytrium* sp., a significant increase was observed with treatment A3. In terms of kg/m^3^ per year, the production was approximately 5953.15 kg/1000 m^3^ per year. To achieve the equivalent squalene yield within a year of production, it would be necessary to capture 15,666.18 Portuguese dogfish sharks. In contrast, obtaining the same amount of squalene from the more squalene-concentrated Birdbeak dogfish would require the capture of 8553.38 sharks (Figure 6).

Considering this, analyzing the production capacity of a biotechnology industry with a volume of 10,000.00 m^3^, it can produce 59.50 t of squalene over one year. However, using traditional methods would entail the capture of approximately 156,661.84 sharks. Although these figures may seem illogical, Jeffreys [27] estimated that 100 million sharks are captured annually by all fishing boats worldwide. Similarly, using the Global Fishing Watch tool, we observed that fishing activity predominantly involves trawl fisheries rather than longline fishing. This information is crucial because such fishing activities contribute to capturing several highly threatened species [28]. This overfishing has caused an ecological imbalance and a major crisis due to inadequate control and monitoring [29].

Another crucial factor when sourcing squalene through a clean biotechnological route is the elimination of impurities in the raw material. Studies have evaluated mercury bioaccumulation in various Australian shark species [30], along with other metals such as arsenic, cadmium, and zinc. Therefore, this in silico sensitivity analysis highlights the importance of changing the source of the raw material.

## 3. Discussion

The results underscore the importance of optimizing cultivation systems for *Thraustochytrium* sp. and *Aurantiochytrium* sp. to achieve sustainable squalene production. *Thraustochytrium* sp. was notably responsive to nitrogen levels, with reduced nitrogen significantly decreasing both biomass and squalene yields, consistent with observations by [18,31], who emphasized nitrogen’s critical role in nucleic acid and protein synthesis. Additionally, considering the previously reported lag phase of this strain, a short lag phase in microbial growth is highly recommended to achieve high productivity, particularly in industrial-scale production [32].

In this strain, glucose is known for its positive effects on biomass production [33]. However, excessively high glucose concentrations can negatively impact biomass production, as observed in the T2 treatment. Gupta et al. [33] demonstrated a decrease in biomass growth when the carbon source was increased to 40 g/L (*w*/*v*). Previous studies on the same strain have also shown that the conversion of carbon source to biomass declines as the initial glucose concentration rises, likely due to growth inhibition caused by high glucose levels [16,34]. Thus, for treatments T2, T3, and T4, lower biomass production was observed, similar to findings by Burja et al. [18], who noted that a high C–N ratio can hinder the production of nitrogen-containing compounds such as proteins and nucleic acids. Nitrogen, a crucial nutrient for algal metabolism, is observed to be deficient under stress conditions with reduced nitrogen, resulting in an excess of carbon source, which is often related to lipid accumulation [35,36].

In contrast, *Aurantiochytrium* sp. thrived under high C–N ratios, exhibiting increased biomass and squalene production under nutrient stress. The strain’s performance under batch conditions suggests potential for commercial scalability. Patel et al. [37] and Martins et al. [1] similarly reported robust growth under optimized C–N ratios, corroborating this study’s findings. They showed a maximum cell concentration of 11.24 g/L achieved by cultivating *Aurantiochytrium* sp. T66 (ATCC PRA-276) with a C–N ratio of 10. Another study by Martins et al. [1] using *Aurantiochytrium* sp. ATCC PRA-276 reported a biomass production of 5.04 g/L (0.07 g/L·h in 72 h) with a C–N ratio similar to treatment A2. In their research, Furlan et al. [15] found that low C–N (<54.50) ratios inhibited the growth of a particular strain, whereas a higher ratio resulted in greater final biomass levels. This implies that *Aurantiochytrium* sp. requires less nitrogen than other Thraustochytrids, such as *Thraustochytrium* sp. [34].

Regarding squalene productivity, extending the growth period for *Aurantiochytrium* sp. to 96 h negatively impacted squalene accumulation (mg/g), which decreased in all trials after 48 h. Researchers have also reported a maximum squalene content followed by a decline as the culture aged, likely due to its conversion into other compounds [1,35,38]. Lewis [39] also demonstrated that squalene content decreased in aged cultures of Thraustochytrid ACEM 6063. Jiang et al. [35] obtained a squalene content of 162 mg/g in dry cell weight and 1.31 mg/L from *Schizochytrium mangrovei* FB1. Squalene is an important intermediate in sterol biosynthesis, with its reduction possibly due to its consumption by the organism or conversion to compounds like sterols via reaction with molecular oxygen, as this reduction was noted with increased culture time under aerobic conditions [34]. Thus, the enzyme squalene monooxygenase requires oxygen molecules to catalyze the oxidation of squalene for sterol biosynthesis, likely being a primary reason for the observed decrease [3,37].

In a previous study, Patel [36] reported the production of biomass with the highest squalene content of 88.47 mg/g and 1.00 mg/L in bioreactor cultivation with an initial glucose concentration of 30 g/L. However, the highest recorded squalene yield was 198 mg/g with a concentration of 1290 mg/L [40]. Additionally, research by Nakazawa [41] suggested that the genus *Aurantiochytrium* sp. may have evolved to accumulate squalene. In comparison with other oleaginous microorganisms such as *Aurantiochytrium* sp. 4W-1b, *Aurantiochytrium* sp. KRS101, and *Aurantiochytrium* sp. P5/2, the highest biomass concentrations were 9.01 g/L [42], 5.5 g/L [43], and 17.43 g/L [42], respectively. Therefore, the higher biomass concentration obtained from *Aurantiochytrium* sp. ATCC PRA 276 supports its commercial application in the production of bioactive compounds.

The higher biomass concentration achieved by *Aurantiochytrium* sp. ATCC PRA 276 demonstrates its feasibility for commercial applications in bioactive compound production. Moreover, replacing shark-derived squalene with microbial sources such as *Thraustochytrium* sp. and *Aurantiochytrium* sp. not only meets industrial squalene demands but also significantly reduces the ecological impact of shark fishing. This transition eliminates impurities such as mercury, arsenic, and cadmium [26,30], addressing the contamination issues inherent in shark-based squalene.

The sensitivity analysis conducted in this study offers critical insights into the sustainability and feasibility of replacing traditional shark-derived squalene with biotechnological alternatives. By examining the relationship between carbon and nitrogen concentrations in different culture systems, we can evaluate the impact of process optimization on industrial squalene production. The analysis emphasizes that varying these parameters significantly influences squalene yields, ultimately affecting the overall productivity of the system.

A key takeaway from this sensitivity analysis is the significant disparity between biotechnological squalene production and the requirement for shark-derived squalene to meet identical market demands. The findings suggest that utilizing oleaginous strains (e.g., *Thraustochytrium* sp. and *Aurantiochytrium* sp.) can reduce the excessive dependence on deep-sea sharks, many of which are already under conservation concerns due to overfishing [17]. Comparing the number of sharks needed to obtain equivalent squalene quantities further underscores the urgency of shifting toward alternative sources. Moreover, this study highlights that biotechnological methods provide a predictable and scalable production paradigm, in contrast to the unpredictability of environmental and economic factors impacting shark-derived squalene production. With climate change and uncontrolled fishing practices exacerbating the stress on marine ecosystems, extracting squalene from sharks is becoming increasingly unsustainable. Meeting the global squalene demand of ~2500 t per year [44] (accessed on 22February 2025) through microbial fermentation is both technically feasible and industrially scalable, with *Aurantiochytrium* sp. as an example, as it had the highest production yield among the strains evaluated. Based on the estimated production capacity of 59.5 t per year and 10,000 m^3^ bioreactor, approximately 42 reactors with 10,000 m^3^ each would be required to meet this demand. This highlights the efficiency of large-scale microbial fermentation in producing squalene on an industrial scale and offers a viable alternative to shark-derived sources. This transition could significantly reduce the exploitation of endangered shark species and is in line with global initiatives for environmental protection and sustainability in the pharmaceutical, cosmetic, and nutraceutical industries. Although *Aurantiochytrium* sp. was used as a reference in this analysis due to its superior productivity, the integration of other high-yielding strains could further increase the efficiency of the bioprocess. In addition, the risk of contamination with heavy metals such as mercury, arsenic, and cadmium in shark-based products raises further concerns regarding their safety and suitability for pharmaceutical and cosmetic purposes. By incorporating this sensitivity analysis into industrial decision-making processes, it becomes clear that switching to microbial squalene production is not only an environmentally responsible decision but also a financially viable strategy. The analysis also shows that refinement of culture conditions, including optimization of carbon and nitrogen supply in fed-batch systems, could significantly improve production efficiency. This advance makes biotechnology a more competitive alternative to the traditional method of extracting squalene from sharks. In summary, the sensitivity analysis provides compelling evidence for the necessity and feasibility of replacing shark-derived squalene with microbial fermentation and demonstrates the potential for a more sustainable and ethical industrial route.

The economic analysis of squalene production by *Aurantiochytrium* sp. using parameters based on the Russo et al. [45] study shows a high financial feasibility with a net present value (NPV) of USD 36.2 million over a 10-year period. The valuation took into account an initial investment of USD 500,000, a discount rate of 8% based on biotechnology projects, and annual operating costs of USD 480,288, including raw materials, energy, labor, and maintenance (Appendix A). The projected revenues, based on an annual production of 59.5 t of squalene and an average price of 100 USD/kg, resulted in an annual cash flow of USD 5.47 million, confirming the economic attractiveness of the bioproduction of squalene. The introduction of this biotechnological process not only enables the economic replacement of animal-derived squalene, reducing shark fishing, but also offers advantages in terms of purity, quality control, and industrial scalability, making it a sustainable and competitive solution for the cosmetic, pharmaceutical, and nutraceutical industries. Future research should aim at scaling up these cultivation systems, investigating alternative carbon and nitrogen sources, and boosting yields through genetic and metabolic engineering. Such efforts will affirm microbial squalene’s status as a sustainable alternative, promoting ecosystem balance and reducing dependence on marine resources.

## 4. Materials and Methods

### 4.1. Microorganisms and Cultivation System

Cells from *Thraustochytrium* sp. ATCC^®^ 26185^™^ and *Aurantiochytrium* sp. ATCC^®^ PRA-276^™^ were obtained from the American Type Culture Collection (Manassas, VA, USA). We maintained the cells of both strains on potato dextrose agar at 4 °C until they were needed for inoculum preparation. To prepare the inoculum, we transferred the cells to a 500 mL flask containing 100 mL of culture medium (Table 1). The cells were then incubated in an orbital shaker (KS 260B, IKA, Hong Kong, China) at 150 rpm and 30 °C in the absence of light for 48 h, following the method outlined by Furlan et al. [15].

### 4.2. Culture Conditions

The experiment was segmented into different treatments for each strain and cultured according to a previous study that aimed for the best biomass yield. Cultivations of both strains were conducted in a bench Biostat^®^ BPlus bioreactor (Sartorius Stedim Biotech GmbH, Goettingen, Germany) with a maximum capacity of 5 L. This bioreactor is composed of a borosilicate glass culture vessel and is equipped with pressure flow meters, as well as gas and liquid control units. After each experiment, the bioreactor was sterilized by autoclaving in an autoclave (Uniclave 77-127 L, A. J. Costa Irmãos Lda., Cacém, Portugal) for 60 min. To examine the effects of carbon and nitrogen on *Thraustochytrium* sp. and *Aurantiochytrium* sp., the same basal formula was initially used (Table 2).

In this study, glucose served as the carbon source for both *Thraustochytrium* sp. and *Aurantiochytrium* sp. strains. The nitrogen sources for *Thraustochytrium* sp. included ammonium sulfate and yeast extract. For *Aurantiochytrium* sp., ammonium sulfate, yeast extract, and monosodium glutamate were used. We applied different treatments to each strain to evaluate the effects of various carbon and total nitrogen concentrations on squalene production in both batch and fed-batch systems. The control treatments were designated as T1 for *Thraustochytrium* sp. and A1 for *Aurantiochytrium* sp. Table 2 provides a clear and concise summary of the experimental conditions for both strains. We sterilized all dissolved components by filtration through a 0.22 µm Millipore membrane (EMD Millipore, Billerica, MA, USA), except for yeast extract, monosodium glutamate, and glucose solutions, which we separately sterilized by autoclaving at 121 °C for 15 min in an autoclave (CV-EL-18 L, Certoclav GmbH, Traun, Austria).

We added 350 mL from each inoculum to the culture medium (10%, *v/v* inoculum volume/total culture medium volume). The treatments were maintained at 23 °C with an agitation of 100 rpm and a pH of 6.0, adjusted with 4 N NaOH. We controlled the dissolved oxygen concentration in the medium at 5% saturation through aeration (0–2.5 vvm). The experiments for both strains concluded after a 96 h cultivation period. We collected cells on days 24, 48, 72, and 96 for biochemical component determination. Each group had two biological replicates, and each assay included three technical replicates.

### 4.3. Productivity Parameters

Biomass concentration was expressed in terms of dry weight, which was measured at 24 h intervals as per Min et al. [46]. Glucose concentration in the culture medium’s supernatant was determined every 24 h using the spectrophotometric method proposed by Miller [45]. Meanwhile, the total nitrogen content, including both defined and complex sources, was measured every 24 h following the procedure described by Furlan et al. [15].

### 4.4. Squalene Content Determination

Squalene was quantified in each collected fraction following centrifugation of the cells at 8742× *g* for 15 min at 4 °C. The biomass was then washed with distilled water and centrifuged again; this process was repeated twice. After washing, the biomass was frozen at −20 °C and dried for 48 h via lyophilization (Heto PowerDry LL3000 Freeze Dryer, Thermo Fisher Scientific, Bremen, Germany). The quantification of squalene in the freeze-dried biomass cells (100 mg) was performed according to the method described by Chen et al. [13]. Squalene was quantified and identified by comparison with a commercial standard (Sigma-Aldrich, St. Louis, MO, USA).

### 4.5. Squalene Production Estimate

The annual production of squalene was estimated for medium and large industries, each with a daily capacity of 1000–10,000, operating 24 h a day, 365 days a year, as described by Fagundes et al. [5]. The fisheries study area was assessed, focusing on sharks, to compare with the biotechnological potential of Thraustochytrids strains from regions in Oceania, between Australia and New Zealand. This choice was informed by the high export volume of shark liver oil from these locations, which serve as significant global sources of raw squalene. The tools employed to define the study area were in silico, sourced from the OCEASEARCH website (www.ocearch.org), which monitors shark species and their migratory patterns, addressing the information gap on fisheries [47]. Furthermore, fishing activities in the region were examined via the Global Fishing Watch website (www.globalfishingwatch.org). For comparative analysis with sharks, the highest squalene productivity treatments were selected: T5 for *Thraustochytrium* sp. and A3 for *Aurantiochytrium* sp. The endangered sharks studied in this region included Bramble shark (*Echinorhinus brucus*)**,** Portuguese dogfish (*Centroscymnus coelolepis*), Birdbeak dogfish (*Deania calceus*), Kitefin shark (*Dalatias licha*), and Leafscale gulper shark (*Centrophorus squamosus*). The economic analysis of squalene production by *Aurantiochytrium* sp. was carried out using a techno-economic valuation framework incorporating the key financial parameters from the Russo et al. [45] study and Lozano-Grande [48]. The valuation was performed using the discounted cash flow (DCF) method, in which the net present value (NPV) was calculated over a 10-year period to assess the feasibility of production on an industrial scale. The analysis considered an initial capital investment (CAPEX) of USD 500,000, reflecting the estimated cost of infrastructure and equipment for a fermentation-based production facility. The operating expenses (OPEX), which amounted to USD 480,288 per year, included the cost of raw materials (glucose, ammonium sulfate, yeast extract), energy consumption, labor, and maintenance. To ensure a realistic and optimal economic assessment, the best performing growing condition was selected based on the highest squalene yield to maximize productivity and cost efficiency. The income stream was determined based on an annual production capacity of 59.5 t of squalene using the highest yield conditions reported for *Aurantiochytrium* sp. and an average market price of 100 USD/kg. The resulting annual cash flow of USD 5.47 million was discounted at a rate of 8%, a standard value in biotechnology investment valuations, to arrive at the net present value of USD 36.2 million. This analysis provides a comprehensive understanding of the financial viability of squalene production through microbial fermentation and underpins its potential as a sustainable and economically attractive alternative to traditional shark-derived sources.

### 4.6. Statistical Analysis

We conducted an analysis of variance (general linear model, one-way analysis of variance) using Statistica^®^ software (version 6.1, StatSoft, Tulsa, OK, USA). To determine the differences in means between pairs, we employed confidence intervals in a Tukey HSD test, setting the significance level at *p* < 0.05.

## 5. Conclusions

For *Thraustochytrium* sp. cultures, squalene yield (mg/g) decreased with increases in the initial glucose and nitrogen concentrations. A restricted nitrogen supply in *Aurantiochytrium* sp. batch cultivation led to an increase in squalene production. The highest squalene concentration, 54.97 mg/L, was achieved after 96 h of *Aurantiochytrium* sp. cultivation, utilizing initial concentrations of 30 g/L glucose and 0.22 g/L nitrogen in a batch culture system. Therefore, in analyzing a biotechnology industry with a capacity of 10,000 m^3^/year, it is feasible to produce 59.50 t of squalene. Nevertheless, employing traditional methods would require the annual capture of approximately 156,661 sharks.

## Figures and Tables

**Figure 1 marinedrugs-23-00132-f001:**
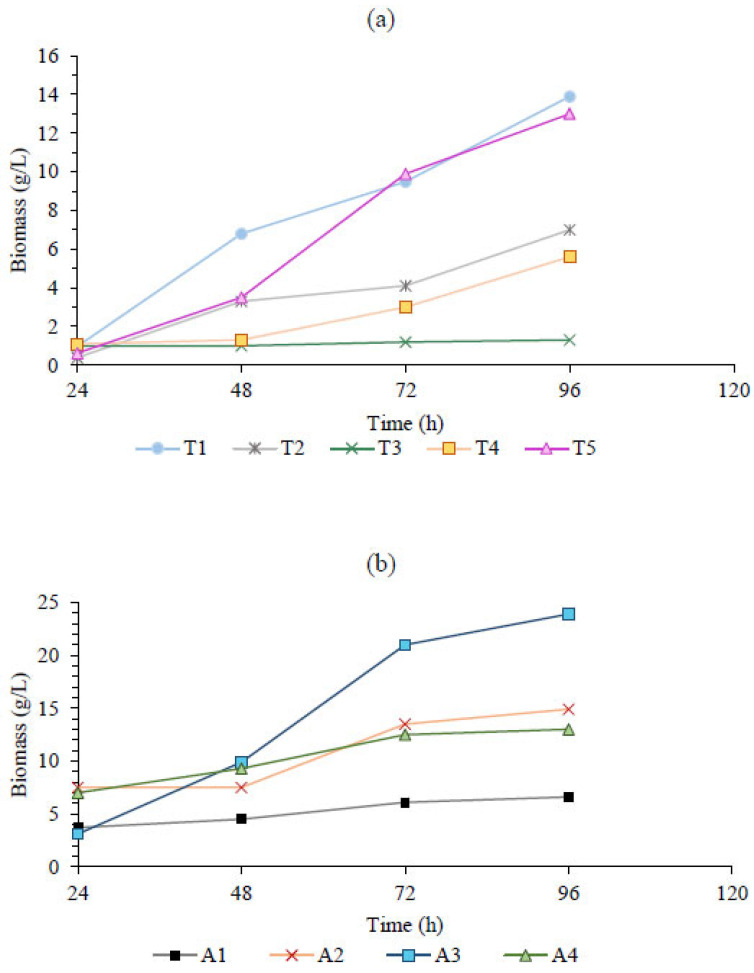
Culture conditions and biomass production of (**a**) *Thraustochytrium* sp. after the treatments (g/L): (T1) Batch system, glucose: 30, nitrogen: 2.4, carbon/nitrogen: 5; (T2) batch system, glucose: 60, nitrogen: 2.4, carbon/nitrogen: 10; (T3): batch system, glucose: 30, nitrogen: 0.8, carbon/nitrogen: 15; (T4) fed-batch system, glucose: 30, nitrogen: 0.009 per hour, carbon/nitrogen: 13.9; (T5) fed-batch system, glucose: 0.1 per hour, nitrogen: 2.4, carbon/nitrogen: 1.6, and of (**b**) *Aurantiochytrium* sp. after the treatments (g/L): (A1) batch system, glucose: 30, nitrogen: 3, carbon/nitrogen: 4; (A2) batch system, glucose: 30 nitrogen: 0.44, carbon/nitrogen: 27.2; (A3) batch system, glucose: 30, nitrogen: 0.22, carbon/nitrogen: 54.5; (A4) fed-batch system, glucose: 0.14 per hour, nitrogen: 0.0014 per hour, carbon/nitrogen: 40.

**Figure 2 marinedrugs-23-00132-f002:**
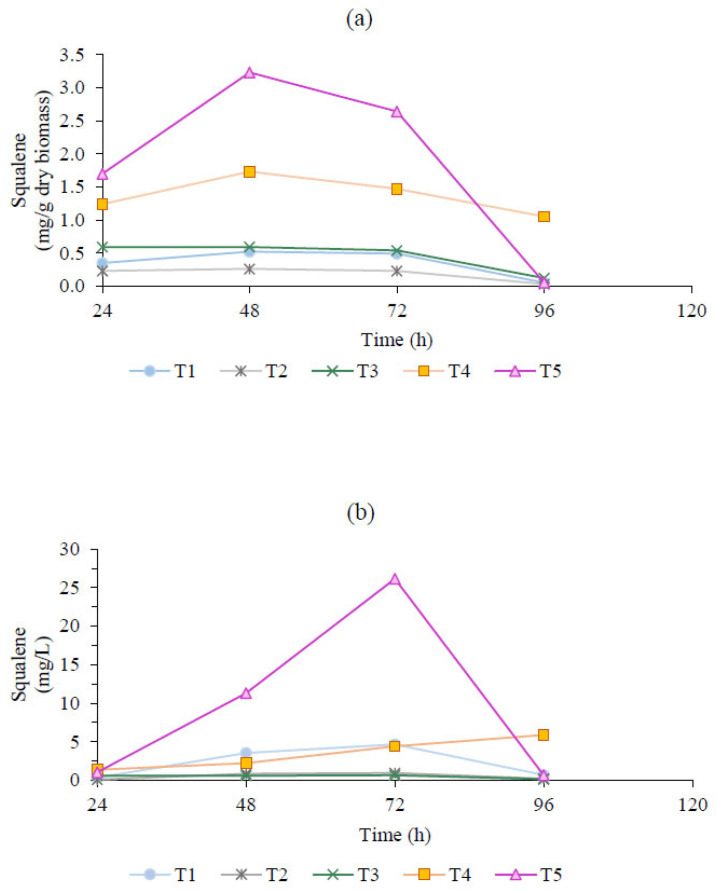
The squalene yield (mg/g) (**a**) and concentration (mg/L) (**b**) under different culture conditions from *Thraustochytrium* sp. The treatments were as follows (g/L): (T1) batch system, glucose: 30, nitrogen: 2.4, carbon/nitrogen: 5; (T2) batch system, glucose: 60, nitrogen: 2.4, carbon/nitrogen: 10; (T3) batch system, glucose: 30, nitrogen: 0.8, carbon/nitrogen: 15; (T4) fed-batch system, glucose: 30, nitrogen: 0.009 per hour, carbon/nitrogen: 13.9; (T5) fed-batch system, glucose: 0.1 per hour, nitrogen: 2.4, carbon/nitrogen: 1.6.

**Figure 3 marinedrugs-23-00132-f003:**
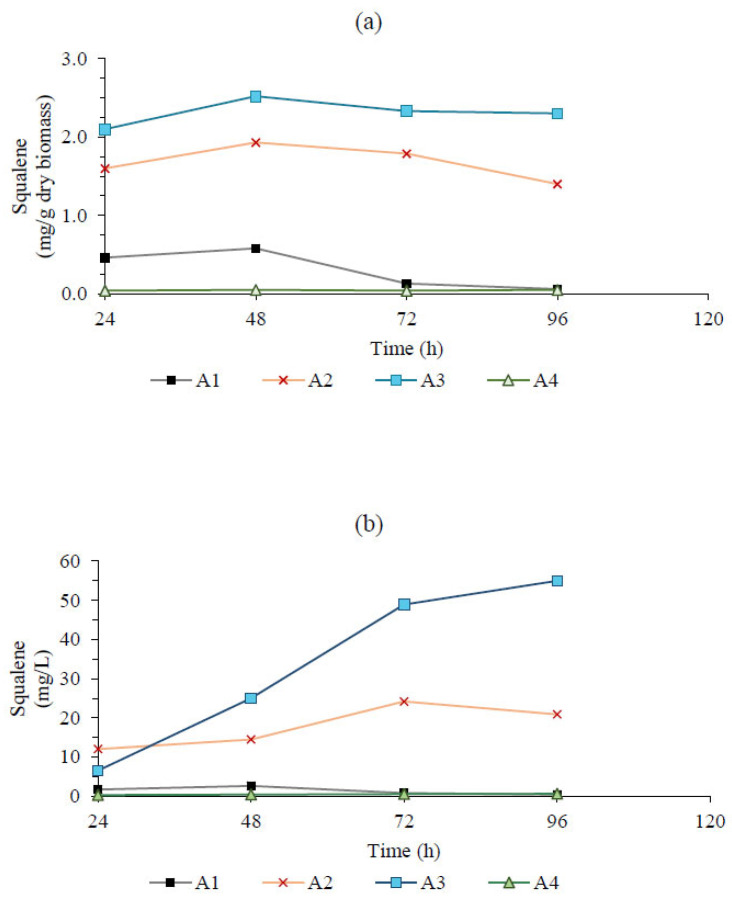
The squalene yield (mg/g) (**a**) and concentration (mg/L) (**b**) under different culture conditions from *Aurantiochytrium* sp., showing the following treatments (g/L): (A1) batch system, glucose: 30, nitrogen: 3, carbon/nitrogen: 4; (A2) batch system, glucose: 30 nitrogen: 0.44, carbon/nitrogen: 27.2; (A3) batch system, glucose: 30, nitrogen: 0.22, carbon/nitrogen: 54.5; (A4) fed-batch system, glucose: 0.14 per hour, nitrogen: 0.0014 per hour, carbon/nitrogen: 40.

**Figure 4 marinedrugs-23-00132-f004:**
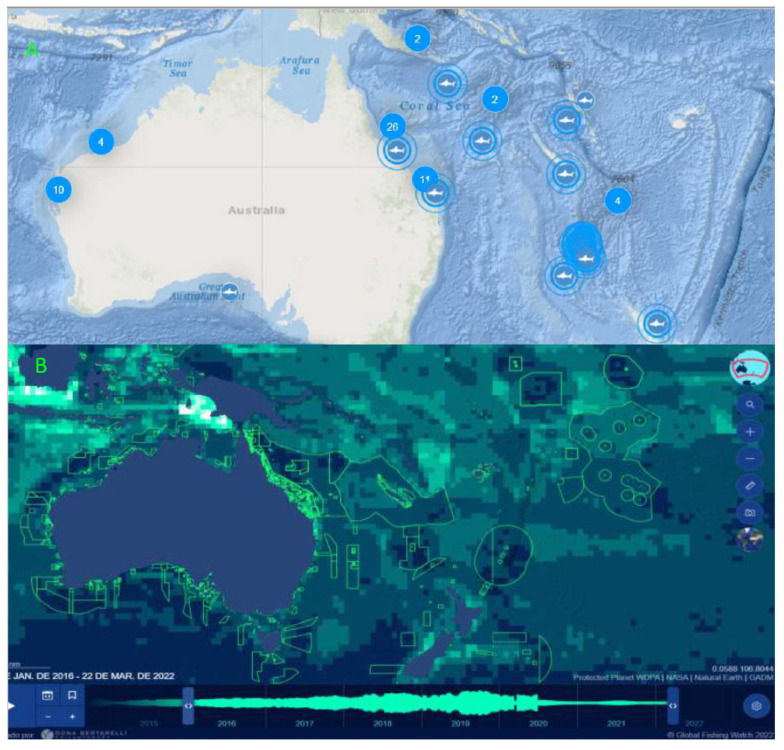
Evaluation of the study area: (**A**) shark activity and (**B**) fishing activity in Oceania. The green lines indicate conservation areas.

**Figure 5 marinedrugs-23-00132-f005:**
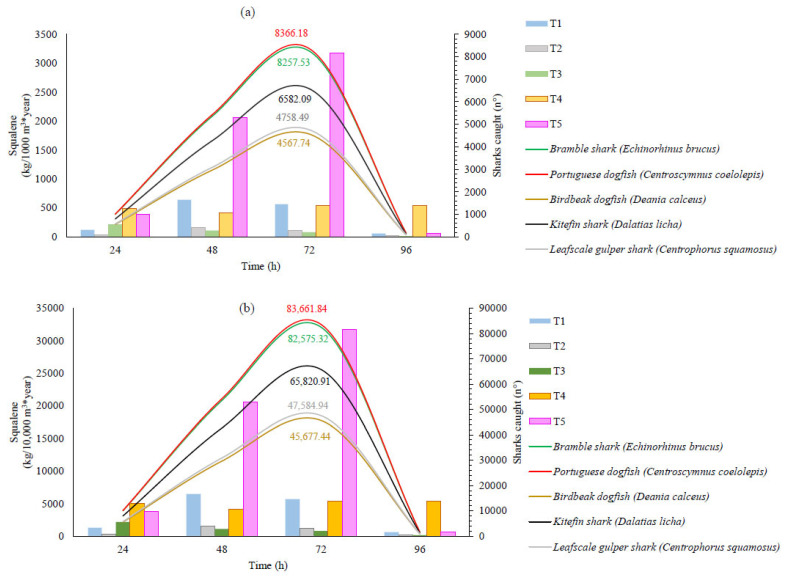
Squalene sensitivity analysis applied to each culture treatment of *Thraustochytrium* sp. in an industrial setting with medium (**a**) and high capacity (**b**). This analysis is compared to the estimated number of sharks needed to produce the same amount of squalene. The estimation was conducted by considering the squalene content from one animal of each species. The treatments were as follows (g/L): (T1) batch system, glucose: 30, nitrogen: 2.4, carbon/nitrogen: 5; (T2) batch system, glucose: 60, nitrogen: 2.4, carbon/nitrogen: 10; (T3) batch system, glucose: 30, nitrogen: 0.8, carbon/nitrogen: 15; (T4) fed-batch system, glucose: 30, nitrogen: 0.009 per hour, carbon/nitrogen: 13.9; (T5) fed-batch system, glucose: 0.1 per hour, nitrogen: 2.4, carbon/nitrogen: 1.6.

**Figure 6 marinedrugs-23-00132-f006:**
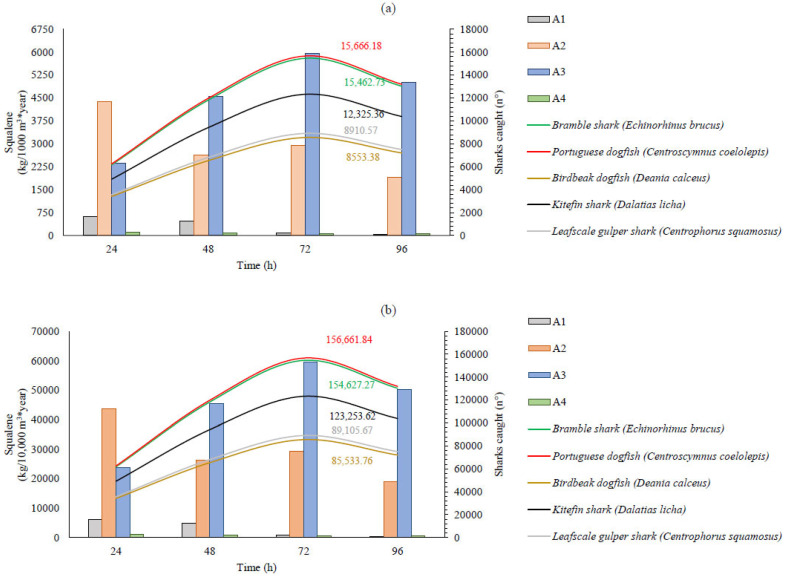
A squalene sensitivity analysis was conducted for each culture treatment of *Aurantiochytrium* sp. production in an industry with medium (**a**) and high capacity (**b**). The results are compared with the number of sharks required to obtain an equivalent amount of squalene. The estimate was calculated based on the squalene content derived from one animal of each species. The treatments, measured in grams per liter (g/L), were as follows: (A1) batch system, glucose: 30, nitrogen: 3, carbon/nitrogen: 4; (A2) batch system, glucose: 30, nitrogen: 0.44, carbon/nitrogen: 27.2; (A3) batch system, glucose: 30, nitrogen: 0.22, carbon/nitrogen: 54.5; (A4) fed-batch system, glucose: 0.14 per hour, nitrogen: 0.0014 per hour, carbon/nitrogen: 40.

**Table 1 marinedrugs-23-00132-t001:** Inoculum preparation for both strains.

Variables	*Thraustochytrium* sp.	*Aurantiochytrium* sp.
Yeast extract (g/L)	1	1
Peptone (g/L)	1	15
Glucose (g/L) *	5	20
Seawater (*w*/*v*)	1.50%	1.50%

* The glucose stock solution was sterilized separately, for both strains.

**Table 2 marinedrugs-23-00132-t002:** Experimental culture conditions for both strains.

*Thraustochytrium* sp.						*Aurantiochytrium* sp.				
Components from Basal Medium (mg/L)						Components from Basal Medium (mg/L)				
KH_2_PO_4_	1540					KH_2_PO_4_	300			
MgSO_4_·7H_2_O	2620					MgSO_4_·7H_2_O	5000			
NaCl	710					NaCl	10,000			
NaHCO_3_	-					NaHCO_3_	100			
CaCl_2_·2H_2_O	-					CaCl_2_·2H_2_O	300			
KCl	-					KCl	280			
Dissolved components (mg/L)						Dissolved components (mg/L)				
MnCl_2_·4H_2_O	3.00					MnCl_2_·4H_2_O	8.60			
ZnSO_4_·7H_2_O	3.00					ZnCl_2_	0.60			
CoCl_2_·6H_2_O	0.04					CoCl_2_·4H_2_O	0.26			
Na_2_MoO_4_·2H_2_O	0.04					CuSO_4_·5H_2_O	0.02			
CuSO_4_·5H_2_O	2.00					FeCl_3_·6H_2_O	2.90			
NiSO_4_·6H_2_O	2.00					H_3_BO_3_	34.20			
FeSO_4_·7H_2_O	10.00					Na_2_EDTA	30.00			
Thiamine	9.50					Thiamine	9.50			
Calcium pantothenate	3.20					Calcium pantothenate	3.20			
Treatment	T1 (C)	T2	T3	T4	T5	Treatment	A1 (C)	A2	A3	A4
C/N	5	10	15	13.9	1.6	C/N	4	27.2	54.5	40
Glucose (g/L)	30	60	30	30	0.1 *	Glucose (g/L)	30	30	30	0.14 *
Total nitrogen (g/L)	2.4	2.4	0.8	0.009 *	2.4	Total nitrogen (g/L)	3	0.44	0.22	0.0014 *
(NH_4_)_2_SO_4_ (g/L)	6.25	6.25	1.89	0.021 *	6.3	(NH_4_)_2_SO_4_ (g/L)	1.36	0.20	0.1	0.00066 *
Yeast extract (g/L)	8.8	8.8	3.23	0.036 *	8.8	Yeast extract (g/L)	13.63	2	1	0.0066 *
Monosodium glutamate (g/L)	-	-	-	-	-	Monosodium glutamate (g/L)	13.63	2	1	0.0066 *

C: Control; * g/L·h (fed-batch).

## Data Availability

The data presented in this study are available on request from the corresponding author.

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
