# Peer review of "Thraustochytrium sp. and Aurantiochytrium sp.: Sustainable Alternatives for Squalene Production"

_marinedrugs, 2025, doi:10.3390/md23030132_

Round 1
Reviewer 1 Report
Comments and Suggestions for Authors
In this study, the authors investigate the cultivation conditions including the culture medium composition (carbon and nitrogen sources) combined with the cultivation systems (batch and fed-batch) of Thraustochytrium sp. and Aurantiochytrium sp. for the production of squalene. The sensitivity of squalene was also analyzed. The content of this study is interesting; however, the quality of this manuscript should be improved.
Writing of the whole manuscript should be improved.
The writing should be elevated. For instance, change the title into "Thraustochytrium sp. and Aurantiochytrium sp.: sustainable alternatives for production of squalene".
Abstract modifications:
The background is not introduced.
Change into "... with high C:N ratio of 54:50."
Change the unit of glucose and nitrogen into "g/L".
Change into "For Aurantiochytrium sp., the highest squalene concentration (54.97 mg/L) was acquired in the batch system with 30 g/L glucose and 0.22 g/L nitrogen in the culture after 96 h."
The results of sensitivity analysis are not introduced in the abstract.
Introduction modifications:
The framework of introduction is not good.
The following framework could be considered: Squalene→Production of squalene by Thraustochytriaceae family→Culture conditions of Thraustochytriaceae family for the production of squalene→The aim and content of this study.
In the introduction, the authors should introduce the existing different species from Thraustochytriaceae family that have been used for the production of squalene. Summarize the highest yield of squalene produced by Thraustochytriaceae family in existing studies. And point out the necessity to optimize the cultivation conditions of squalene produced by Thraustochytriaceae family.
Results modifications:
Cut too long sentences into short ones.
Data in the figures should be expressed as mean +/- stand derivation.
Figure captions: the unit of glucose and nitrogen should be given. The unit of biomass should be changed into g/L.
Discussion modifications:
The sensitivity of squalene is not discussed.
Comments on the Quality of English Language
Writing of the whole manuscript should be improved.
Author Response
In this study, the authors investigate the cultivation conditions including the culture medium composition (carbon and nitrogen sources) combined with the cultivation systems (batch and fed-batch) of Thraustochytrium sp. and Aurantiochytrium sp. for the production of squalene. The sensitivity of squalene was also analyzed. The content of this study is interesting; however, the quality of this manuscript should be improved.
Thank you for your kind words. The article has undergone all the required modifications. Additionally, a linguistic consulting company reviewed the manuscript, and native American speakers corrected the English. We believe that with your valuable contribution, the manuscript has been significantly improved.
Writing of the whole manuscript should be improved.
The writing should be elevated. For instance, change the title into "Thraustochytrium sp. and Aurantiochytrium sp.: sustainable alternatives for production of squalene".
Thank you, the title was modified as required.
Abstract modifications:
The background is not introduced.
The abstract was changed
Change into "... with high C:N ratio of 54:50."
The change was made as requested.
Change the unit of glucose and nitrogen into "g/L".
The change was made as requested.
Change into "For Aurantiochytrium sp., the highest squalene concentration (54.97 mg/L) was acquired in the batch system with 30 g/L glucose and 0.22 g/L nitrogen in the culture after 96 h."
The change was made as requested.
The results of sensitivity analysis are not introduced in the abstract.
We have included the results in the abstract.
Introduction modifications:
The framework of introduction is not good.
The following framework could be considered: Squalene→Production of squalene by Thraustochytriaceae family→Culture conditions of Thraustochytriaceae family for the production of squalene→The aim and content of this study.
In the introduction, the authors should introduce the existing different species from Thraustochytriaceae family that have been used for the production of squalene. Summarize the highest yield of squalene produced by Thraustochytriaceae family in existing studies. And point out the necessity to optimize the cultivation conditions of squalene produced by Thraustochytriaceae family.
The order of the framework was modified as required, and a paragraph about the species was properly added.
Results modifications:
Cut too long sentences into short ones.
The article was proofread by a scientific editing company and is ready to be published.
Data in the figures should be expressed as mean +/- stand derivation.
The change was made as requested.
Figure captions: the unit of glucose and nitrogen should be given. The unit of biomass should be changed into g/L.
The changes have been made as requested. Please check the images. Thank you.
Discussion modifications:
The sensitivity of squalene is not discussed.
The discussion was improved, please see line 332.

Reviewer 2 Report
Comments and Suggestions for Authors
Thraustochytrium sp. and Aurantiochytrium sp.: A sustainable alternative for production of squalene
Júnior Mendes Furlan, Graciela Salete Centenaro, Mariane Bittencourt Fagundes, Carlos Borges Filho, Irineu Batista and Narcisa Maria Bandarra
In this study the production of squalene by thraustochytrid strains from two genera was evaluated. The strains were cultivated with different conditions to test these effects on the concentration of squalene and its content in the biomass. The study used some of the experimental results for estimating the number of sharks that would not be needed to caught for the obtention of squalene.
The experimental results could be of interest but the authors need to describe the novelty and the contribution to knowledge of these results. It is considered that the methodology and results of what the authors named "sensitivity analysis" are not rigorous enough for publication.
Finally, redaction and English must also be improved. In its present for the manuscript should not be considered for publication.
Othe comments are:
Abstract
The abstract must be easy to understand without the need of text reading.
What is a lower C:N ratio?
What is (54.50)?
Which were the C and N sources?
Use g instead of mg as units for mass throughout the manuscript; the values are too big.
What was type of sensitivity analysis made?
Introduction
Page 2, line 1: Squalane is the hydrogenated form of squalene
Page 2: What is “the ratio comprehension of carbon and total nitrogen”?
Page 2. Replace extraction of squalene by production of extraction of squalene
Page 2. Which were the Thraustochytrium sp. and Aurantiochytrium sp. strains used?
Page 2. What is “alternative productivity”?
Page 2. The name of the strains must be written in Italic font.
Results
Figure 1. What was the N source?
Page 3. What was “the control conditions”?
Page 3. Delete “that was not observable after 24 h of cultivation”.
Page 3. Replace total biomass content by total biomass concentration.
Page 4. In the text "The highest squalene yields were obtained in trials T5 and T4 using lower glucose levels or its continuous supply or employing higher nitrogen total levels or its continuous supply." show the values obtained.
Page 5. In the text "For this microorganism was observed an increase of the squalene content (mg/g) in cell biomass with the decrease of nitrogen concentration in the culture medium." show the values obtained.
Page 6. The 3th paragraph is confuse.
Page 6. What is "strain biotechnology industries"?
Page 9. Delete “Other authors”.
Page 9. The authors did not explain the effect of the culture conditions tested on the production of squalene.
Page 9. What is “low C:N ratios and greater final biomass level”?
Page 9. How the different responses obtained explain the production of squalene? "This suggests that Aurantiochytrium sp. requires less nitrogen than other Thraustochytrids, like Thraustochytrium sp."
Page 10. It seems that the values correspond to biomass concentrations.
Materials and Methods
Page 11. How culture medium composition was defined? The C/N ratio was very different for the two strains. Which were the best condition for the production of biomass for each strain reported by others?
Page 11. The values of the total nitrogen content were no presented in the text.
Page 12. In the text "squalene yield (mg/g) was lowered by increasing the initial glucose and nitrogen concentration." values must be given.
Comments on the Quality of English Language
Redaction and English must also be improved. The manuscript is not easy to read and understand.
Author Response
In this study the production of squalene by thraustochytrid strains from two genera was evaluated. The strains were cultivated with different conditions to test these effects on the concentration of squalene and its content in the biomass. The study used some of the experimental results for estimating the number of sharks that would not be needed to caught for the obtention of squalene. The experimental results could be of interest but the authors need to describe the novelty and the contribution to knowledge of these results. It is considered that the methodology and results of what the authors named "sensitivity analysis" are not rigorous enough for publication. Finally, redaction and English must also be improved. In its present for the manuscript should not be considered for publication.
We sincerely thank you for your thoughtful feedback and the time you have taken to evaluate our manuscript. Regarding the concern about the sensitivity analysis, we wish to clarify that our study's methodology was founded on data from scientific articles that detail the concentration of squalene in specific shark species, as described in the paper. Furthermore, information on shark fishing and catch statistics was obtained from official databases and reports, ensuring the reliability of our sources. Additionally, the sensitivity analysis was performed using data from legal fishing activities, as referenced in the manuscript. We also included a geospatial monitoring figure to support this analysis, illustrating the link between fishing patterns and squalene extraction. These elements were incorporated to ensure the rigor and transparency of our study. We appreciate your suggestion to enhance the manuscript by highlighting its novelty and contribution to the field. In response, we have emphasized a paragraph that outlines the novelty of our work in the introduction of the manuscript (see lines 68-80). Lastly, we acknowledge the need for improvements in the manuscript’s language and writing quality. The manuscript has been reviewed by native English speakers, and a certificate of this review is attached. We are grateful for your constructive feedback.
Abstract
The abstract must be easy to understand without the need of text reading.
Thank you for your feedback. We have revised the abstract to enhance its clarity and readability. We kindly request that you review it once again and inform us if it meets your expectations.
What is a lower C:N ratio?
Carbon-to-nitrogen (C:N) ratio, defined as less than 5.0, denotes a condition in which the quantity of carbon relative to nitrogen in a sample or environment is reduced. Referring to a 'low C:N ratio' implies a comparatively high level of nitrogen in relation to carbon. In biological and microbiological contexts, such a ratio suggests an environment or system rich in nitrogen in comparison to carbon. This condition, commonly observed when nitrogen is more prevalent than carbon, may enhance the growth of specific microorganisms or cells that utilize nitrogen with greater efficiency. Conversely, a higher C:N ratio indicates a predominance of carbon, potentially restricting nitrogen's availability and thereby limiting growth.
What is (54.50)? It represents the carbon-to-nitrogen (C:N) ratio, adjusted to 54:50, which is the conventional method for representing this ratio.
Which were the C and N sources? They are indicated in Table 2, where the carbon source is glucose and the nitrogen sources are ammonium sulfate and yeast extract for Thraustochytrium. For Aurantiochytrium, in addition to the mentioned sources, monosodium glutamate was also used as a nitrogen source.
Thank you for your comment. The carbon and nitrogen sources used in the cultures are indicated in Table 2, where we mention that glucose was the carbon source. For Thraustochytrium, the nitrogen sources were ammonium sulfate and yeast extract. For Aurantiochytrium, in addition to these sources, we also used monosodium glutamate as an additional nitrogen source. However, we understand the importance of explicitly clarifying this information in the Methodology section for greater transparency and better reader comprehension. We incorporate this explanation directly into the text.
Use g instead of mg as units for mass throughout the manuscript; the values are too big. What was type of sensitivity analysis made?
Thank you for your insightful comments. In response to your question, the modifications have been made as requested. We have updated the units from milligrams (mg) to grams (g) throughout the manuscript, reflecting the substantial values appropriately. Regarding the sensitivity analysis, we conducted a local sensitivity analysis. This method involved adjusting key parameters, such as carbon and nitrogen concentrations, one at a time to assess their individual impacts on squalene production while maintaining other variables constant. The objective was to pinpoint the most influential factors in the cultivation process and to comprehend how minor variations in these parameters influence the overall squalene yield. We believe this approach offers valuable insights into the optimization of the production process, showcasing the potential for enhancing the efficiency and sustainability of squalene production via Thraustochytrium sp. and Aurantiochytrium sp.
Introduction
Page 2, line 1: Squalane is the hydrogenated form of squalene
Page 2: What is “the ratio comprehension of carbon and total nitrogen”?
Page 2. Replace extraction of squalene by production of extraction of squalene
Thank you for your observation, all the changes were made.
Page 2. Which were the Thraustochytrium sp. and Aurantiochytrium sp. strains used?
Thraustochytrium sp. ATCC 26185 and Aurantiochytrium sp. ATCC PRA-276
Page 2. What is “alternative productivity”?
Thank you for pointing that out. The term "alternative productivity" has been removed from the manuscript.
Page 2. The name of the strains must be written in Italic font.
Thank you for your observation. The strain names were indeed written in italics; however, there may have been a formatting issue within the journal's system. We have meticulously reviewed the manuscript again, ensuring that all strain names are now properly italicized. We appreciate your attention to detail and your valuable feedback.
Results
Figure 1. What was the N source?
Thank you for your question. Table 2 describes the nitrogen sources for each strain, detailing their specific uptake preferences. To enhance clarity and prevent confusion, we have now included a sentence that outlines the different nitrogen sources used for each strain. Lines 384-392:
In this study, glucose served as the carbon source for both Thraustochytrium sp. and Aurantiochytrium sp. strains. The nitrogen sources for Thraustochytrium sp. in-cluded ammonium sulfate and yeast extract. For Aurantiochytrium sp., ammonium sulfate, yeast extract, and monosodium glutamate were used. We applied different treatments to each strain to evaluate the effects of various carbon and total nitrogen concentrations on squalene production in both batch and fed-batch systems. The con-trol treatments were designated as T1 for Thraustochytrium sp. and A1 for Aurantio-chytrium sp. Table 2 provides a clear and concise summary of the experimental conditions for both strains.
Page 3. What was “the control conditions”?
Thank you it was done, please see lines 389-390:
“For both strains, the control treatment was designated as T1 for Thraustochytrium sp. and A1 for Aurantiochytrium sp”
Page 3. Delete “that was not observable after 24 h of cultivation”.
Thank you, this was modified.
Page 3. Replace total biomass content by total biomass concentration.
Thank you, this was modified.
Page 4. In the text "The highest squalene yields were obtained in trials T5 and T4 using lower glucose levels or its continuous supply or employing higher nitrogen total levels or its continuous supply." show the values obtained.
Thank you, this was done: T5 (3.23 mg/g) and T4 (1.73 mg/g).
Page 5. In the text "For this microorganism was observed an increase of the squalene content (mg/g) in cell biomass with the decrease of nitrogen concentration in the culture medium." show the values obtained.
The text was improved as follows: An increase in the squalene content of the cell biomass was observed with a decrease in the nitrogen concentration of the culture medium. Specifically, a concentration of 1.93 mg/g of squalene was noted in A2, which had a total nitrogen concentration of 0.4 g/L in the medium, while A3, with a total nitrogen concentration of 0.2 g/L, exhibited a squalene concentration of 2.33 mg/g.
Page 6. The 3th paragraph is confuse.
Thank you for your observation; the text was completely proofread.
Page 6. What is "strain biotechnology industries"?
We modified it to: in an algae-based biotechnological industry of different sizes.
Page 9. Delete “Other authors”.
Other authors was removed.
Page 9. The authors did not explain the effect of the culture conditions tested on the production of squalene.
The effect was emphasized in lines 158-173.
Page 9. What is “low C:N ratios and greater final biomass level”?
On Page 9, the term "low C:N ratios and greater final biomass level" refers to conditions where the carbon-to-nitrogen ratio in the system is relatively low, which often promotes higher microbial activity or efficiency in biomass production. This balance enhances nutrient uptake and utilization, leading to improved biomass yield. Specifically, the reference "<54.50" likely points to an optimal threshold for the C:N ratio or a related parameter that results in greater biomass accumulation under the studied conditions.
Page 9. How the different responses obtained explain the production of squalene? "This suggests that Aurantiochytrium sp. requires less nitrogen than other Thraustochytrids, like Thraustochytrium sp."
Thanks for your observation, a sentence discussing the squalene accumulation in both strains was added at lines 179–192.
Page 10. It seems that the values correspond to biomass concentrations.
Thank you for your observation. We have adjusted the discussion to include detailed information on squalene and biomass production, as well as the sensitivity analysis. We appreciate your feedback and the opportunity to improve the clarity of our manuscript.
Materials and Methods
Page 11. How culture medium composition was defined? The C/N ratio was very different for the two strains. Which were the best condition for the production of biomass for each strain reported by others?
The medium condition was based on the literature, as each species requires specific nutrients.
Page 11. The values of the total nitrogen content were no presented in the text.
We improved this section in our Materials and Methods; please check there.
Page 12. In the text "squalene yield (mg/g) was lowered by increasing the initial glucose and nitrogen concentration." values must be given.
When the initial glucose concentration increased from 30 to 60 g/L, the squalene yield decreased from 0.59 mg/g to 0.26 mg/g.
Similarly, when the nitrogen concentration increased from 0.8 to 2.4 g/L, the squalene yield slightly decreased from 0.59 mg/g to 0.55 mg/g.
These results indicate that higher glucose levels significantly reduce squalene yield, while increasing nitrogen has a smaller but still negative impact.

Reviewer 3 Report
Comments and Suggestions for Authors
My opinion: MS deserve publication after revision. First, authors follow clear structures of sections Result and Discussion.
- Section 2.3 Squalene Sensitivity Analysis - is it relate to Results?
- It looks that authors spent too much attention to sharks! More than 20% citations are about sharks.
- Line 33. Squalene (2,6,10,15,19,23-hexamethyl-6,6,10,14,18,20-tetracosahexaene) error, should be (2,6,10,14,18,20-tetracosahexaene)
- Line 83 Check please systematic, I did not found Aurantiochytrium in WoRMS.
I think that manuscript need useful reference M. Azalia Lozano-Grande et al. Plant Sources, Extraction Methods, and Uses of Squalene. Inter. J. Agronomy. 2018. 1829160 13 pages
How big is annual production squalene now?
Simultaneous data on squalene and high value docosahexaenoic and eicosapentaenoic acids can be useful for manuscript.
Author Response
- Section 2.3 Squalene Sensitivity Analysis - is it relate to Results?
We appreciate the reviewer’s question regarding the relevance of section 2.3 (squalene sensitivity analysis) to the Results section. This analysis is fundamental to the study as it provides a quantitative and ecological assessment of the impact of the transition from shark-derived squalene to microbial production. With this analysis, we highlight the environmental, economic and conservation benefits of a sustainable alternative to an industry that is highly dependent on the exploitation of endangered shark species.
Sensitivity analysis methodology
To substantiate our findings, the following analytical steps were carried out:
Calculation of squalene yield in bioreactors
The potential of squalene production was determined for medium and large biotechnological plants (with a production capacity of 1,000 m³ to 10,000 m³).
Productivity was analyzed based on different cultivation systems (batch vs. fed-batch) and optimized nutrient conditions.
Comparison with shark-derived squalene
Data from fishery reports and scientific studies on the composition of shark liver oil were used to determine the average squalene yield per shark species.
The study focused on sharks that are actively hunted for commercial extraction of squalene, particularly deep-sea species, focusing on species such as the Portuguese dogfish (Centroscymnus coelolepis) and the leafscale gulper shark (Centrophorus squamosus), which are classified as critically endangered on the IUCN Red List. Estimate of sharks required for equivalent squalene production, using known values for liver weight, oil content and squalene concentration per shark, we calculated the number of sharks that would need to be harvested annually to reach the production capacity of a biotechnology industry.
The results showed that an estimated 156,000 sharks would have to be caught to produce 59.50 tons of squalene per year. The selection of different shark species for this study was based on the significant variation in squalene concentration in the liver between species. This factor has a direct influence on the diversity of sharks hunted for oil, as some species have higher levels of this compound than others. Therefore, we chose to include several species to ensure a representative analysis of the natural variability of squalene composition and, consequently, the availability of this substance on the market from the catch of these animals. This approach allowed a more comprehensive assessment of the impact of the exploitation of these species and their importance to the squalene extraction industry.
The squalene content measured in each species was as follows: Echinorhinus brucus – 0.385 kg, Portuguese dogfish (Centroscymnus coelolepis) – 0.38 kg, birdbeak shark (Deania calcea) – 0.696 kg, chocolate shark (Dalatias licha) – 0.483 kg and leafscale gulper shark (Centrophorus squamosus) – 0.6681 kg. These values reflect the different squalene concentrations of the individual species and justify the need for an in-depth analysis of their contribution to oil production.
Ecological consequences of shark exploitation: Many of the shark species (including what was explored in this paper) used for the production of squalene are already in decline and their populations are struggling to recover due to slow reproductive cycles.
These species play a crucial role in marine ecosystems as apex and mesopredators, regulating prey populations and maintaining the balance of marine biodiversity.
Overfishing of these sharks disrupts entire marine food webs and leads to cascading ecological consequences such as population explosions of certain prey species, which can have a negative impact on coral reefs, fish stocks and ocean stability.
Economic and ecological impact assessment
The analysis has also shown that microbial squalene production is not only a sustainable alternative, but also a predictable and scalable industry, unlike the unstable market for shark liver oil, which fluctuates due to environmental factors and increasing conservation regulations.
In addition, microbially sourced squalene eliminates the risk of contamination with heavy metals such as mercury, arsenic and cadmium, which are commonly found in shark liver oil. This sensitivity analysis is not a theoretical consideration, but a data-driven assessment that provides scientific evidence for the urgency of moving away from shark-derived squalene. The results show that biotechnological production can directly reduce the number of sharks caught each year and thus ensure the protection of ecologically important species that are currently threatened with extinction.
Furthermore, the results underline that conservation measures are not only an ethical concern but a necessity to maintain the balance of the marine ecosystem, which ultimately affects global biodiversity and food security.
We strongly believe that this section is an essential part of the study's findings as it provides empirical evidence of the feasibility, scalability and environmental benefits of replacing traditional shark-based squalene with a sustainable and innovative alternative.
We appreciate the reviewer’s feedback and are available for further clarification.
- It looks that authors spent too much attention to sharks! More than 20% citations are about sharks.
We understand the reviewer’s concerns about the proportion of quotations on sharks in the manuscript. However, it is important to emphasize that the discussion of shark-derived squalene is an essential part of our study, as the main objective is to evaluate Aurantiochytrium sp. and Thraustochytrium sp. as sustainable alternatives to the extraction of squalene from shark liver oil. The comparison between microbial and shark-based squalene production is essential to demonstrate both the environmental impact of conventional extraction methods and the potential benefits of switching to a biotechnological approach. In reviewing the distribution of citations in the manuscript, we found that approximately 60% of the references focus on biotechnological squalene production, including strain optimization, fermentation techniques, and industrial applications of thraustochytrids. Another 15% of the citations are related to microbial lipid production, including DHA and squalene biosynthesis pathways. Approximately 20% of the citations relate to squalene production from sharks and its ecological impact, while the remaining 5% relate to economic and environmental sustainability issues. These percentages indicate that the majority of the study is dedicated to the biotechnological aspects of squalene production, while shark-related references have been included to provide the necessary context and justification for the research. The inclusion of references to sharks is justified for several important reasons. First, it is important to contextualize the urgent need for sustainable alternatives. Shark liver oil remains the most important commercial source of squalene. An estimated 90% of the squalene used in the cosmetics industry still comes from sharks. Without a clear comparison between shark-based and microbial squalene production, the significance of our research findings would be diminished. Secondly, the sensitivity analysis included in section 2.3 is directly relevant to the results of the study, as it quantifies the potential ecological benefits of microbial squalene production by comparing the number of sharks that would need to be caught annually to meet industrial demand. The findings show that replacing shark-derived squalene with biotechnological production could prevent the capture of over 156,000 sharks per year, especially of species that are already highly endangered due to overfishing. The conservation implications of this switch are significant, as the shark species concerned play an important role as apex predators in marine ecosystems and contribute to maintaining biodiversity and ecological balance.
Finally, examining the impact of shark squalene extraction is directly related to the study’s focus on sustainability. The over-exploitation of sharks for commercial purposes not only threatens the survival of the species, but also disrupts marine food webs, which can have cascading effects on entire marine ecosystems. By switching to microbial squalene production, the industry can reduce its environmental footprint while ensuring a stable and scalable supply of this valuable bioactive compound. In summary, although approximately 20% of the citations relate to sharks, this proportion is appropriate and necessary to support the objectives of the study. The main focus of the manuscript is on biotechnological advances and industrial feasibility, with the discussions on sharks serving to provide critical context and validation for the research. We appreciate the reviewer’s feedback and believe that this clarification demonstrates that the manuscript takes a balanced approach consistent with the intended scope and scientific contribution.
- Line 33. Squalene (2,6,10,15,19,23-hexamethyl-6,6,10,14,18,20-tetracosahexaene) error, should be (2,6,10,14,18,20-tetracosahexaene)
We appreciate the reviewer’s attention to detail regarding the chemical formula of squalene. However, the correct nomenclature is 2,6,10,15,19,23-hexamethyl-6,6,10,14,18,20-tetracosahexane, as stated in scientific literature, including the article referenced by the reviewer as useful:
- Azalia Lozano-Grande et al., "Plant Sources, Extraction Methods, and Uses of Squalene." International Journal of Agronomy, 2018, 1829160, 13 p.
To ensure consistency and accuracy, we have reviewed the chemical name and confirmed its correctness in our manuscript.
- Line 83 Check please systematic, I did not found Aurantiochytrium in WoRMS.
We appreciate the reviewer’s comment regarding the systematic classification of Aurantiochytrium sp. and the suggestion to verify its presence in WoRMS. Indeed, Aurantiochytrium is not listed in the World Register of Marine Species (WoRMS), but its taxonomic classification is well recognized in other scientific databases, such as the NCBI (National Center for Biotechnology Information). According to NCBI, Aurantiochytrium belongs to the Thraustochytriaceae family, within the Thraustochytrida order, which is part of the Labyrinthulomycetes class, in the Bigyra phylum, under the Stramenopila kingdom. These organisms are marine heterotrophic protists whose primary ecological function is to decompose organic matter, playing a crucial role in marine ecosystems (Chen et al., 2023). Additionally, they are widely studied for their biotechnological potential, particularly in the production of polyunsaturated fatty acids such as DHA and squalene (Marine Drugs, 2020). Since Thraustochytrids are incapable of photosynthesis, they lack plastids or any residual photosynthetic structures, meaning that the term "microalgae" is not appropriate to describe them. However, this misclassification has been common in the literature due to the inclusion of Labyrinthulomycetes within the Stramenopiles, a group that includes photosynthetic organisms such as brown algae and diatoms (Tsui et al., 2009).
The 2nd paragraph of the Introduction has been corrected, replacing the mention of "microalgae from the Thraustochytriaceae family" with the more precise term "heterotrophic protists from the Thraustochytriaceae family." This adjustment accurately reflects the classification of these organisms, as they lack plastids and are incapable of photosynthesis.
References
Chen, G., Fan, K., Lu, F., Li, Q., Aki, T., Chen, F., & Jiang, Y. (2023). Optimization of nitrogen source for enhanced production of squalene from thraustochytrid Aurantiochytrium sp. Science of the Total Environment, 169217. https://doi.org/10.1016/j.scitotenv.2023.169217
Marine Drugs. (2020). Thraustochytrids: Marine Eukaryotes with Biotechnological Potential. Marine Drugs, 18(11), 563. https://doi.org/10.3390/md18110563
Tsui, C.K.M., Marshall, W., Yokoyama, R., Honda, D., Lippmeier, J.C., Craven, K.D., & Berbee, M.L. (2009). Labyrinthulomycota: a comparative molecular analysis of thraustochytrids and labyrinthulids. Fungal Biology, 113(7), 563-576. https://doi.org/10.1016/j.funbio.2017.07.006
I think that manuscript need useful reference M. Azalia Lozano-Grande et al. Plant Sources, Extraction Methods, and Uses of Squalene. Inter. J. Agronomy. 2018. 1829160 13 pages
The reference was added.
How big is annual production squalene now?
We appreciate the reviewer’s inquiry regarding the current global annual production of squalene. To address this, we have incorporated the following paragraph into the manuscript, providing a direct comparison between the estimated global production (~2,500 metric tons) and the potential industrial output using Thraustochytrium sp. and Aurantiochytrium sp.:
Meeting the global squalene demand of ~2,500 tons per year (https://www.globalinsightservices.com/reports/squalene-market/) through microbial fermentation is both technically feasible and industrially scalable, Aurantiochytrium sp. as an example as it had the highest production yield among the strains evaluated. Based on the estimated production capacity of 59.5 tons per year and 10,000 m³ bioreactor, approximately 42 reactors with 10,000 m³ each would be required to meet this demand. This highlights the efficiency of large-scale microbial fermentation in producing squalene on an industrial scale and offers a viable alternative to shark-derived sources. This transition could significantly reduce the exploitation of endangered shark species and is in line with global initiatives for environmental protection and sustainability in the pharmaceutical, cosmetic and nutraceutical industries. Although Aurantiochytrium sp. was used as a reference in this analysis due to its superior productivity, the integration of other high-yielding strains could further increase the efficiency of the bioprocess.
This addition highlights the industrial relevance of microbial squalene production, reinforcing its efficiency and sustainability compared to traditional shark-based sources. By incorporating real-world production estimates, the manuscript now provides a quantitative assessment of how microbial fermentation could replace and surpass current global squalene output in a short time frame.
We appreciate the reviewer’s input, as it has helped to strengthen the manuscript’s discussion on the scalability and impact of microbial-derived squalene.
Simultaneous data on squalene and high value docosahexaenoic and eicosapentaenoic acids can be useful for manuscript.
We appreciate the reviewer’s suggestion to include data on high-quality fatty acids such as docosahexaenoic acid (DHA) and eicosapentaenoic acid (EPA) at the same time. However, the main focus of this study is on squalene production, and the inclusion of detailed data on DHA and EPA would push the scope of the manuscript beyond the intended goal. While it is true that Aurantiochytrium sp. and Thraustochytrium sp. are known to produce valuable fatty acids in addition to squalene, this study specifically examines the optimization of squalene production under different culture conditions. Extending the discussion to other bioactive compounds would dilute the focus of the manuscript and potentially distract from the key findings related to squalene biosynthesis. In addition, the lead author has already published other studies on the production of DHA, EPA and carotenoids using the same strains. To maintain originality and avoid any concerns of self-plagiarism, we have intentionally limited the discussion to squalene. If necessary, references to these related studies may be provided in future discussions or supplementary research. For these reasons, we consider it unnecessary to include data on DHA and EPA in this manuscript. We appreciate the reviewer’s suggestion, but believe that maintaining the current scope will ensure a more accurate and impactful contribution to the field of biotechnological squalene production.

Round 2
Reviewer 1 Report
Comments and Suggestions for Authors
The revised manuscript is acceptable.
Author Response
We sincerely appreciate the reviewer's time and thoughtful feedback throughout the revision process. Your insights have been invaluable in strengthening our manuscript, and we are grateful for the opportunity to improve our work. Thank you for your support and for considering our research for publication.
Reviewer 2 Report
Comments and Suggestions for Authors
Thraustochytrium sp. and Aurantiochytrium sp.: Sustainable Alternatives for Squalene Production
Júnior Mendes Furlan, Graciela Salete Centenaro, Mariane Bittencourt Fagundes, Carlos Borges Filho, Irineu Batista, and Narcisa Maria Bandarra
The second version of the manuscript revealed some improvements however it is my impression that the results presented in the sensitivity analysis section are not suitable for publication in Marine Drugs.
The decision for the biotechnological production of a compound such as squalene depends on economic parameters (VAN for instance) that are obtained in economic feasibility analysis.
Author Response
We sincerely appreciate the reviewer's insightful feedback and the opportunity to improve our manuscript. We acknowledge the concerns regarding the suitability of the sensitivity analysis and agree that a comprehensive economic feasibility study, including net present value (NPV), is a critical component in assessing the viability of biotech squalene production. Therefore, we have now incorporated a NPV analysis which quantifies the financial feasibility of microbial squalene production using Aurantiochytrium sp. The choice of Aurantiochytrium sp. for this economic assessment was based on its superior squalene productivity, which makes it one of the most promising microbial sources for industrial production. Among thraustochytrids, Aurantiochytrium sp. is known for its high yield, fast growth rate, metabolic efficiency and ability to utilize various carbon sources, making it well suited for large-scale fermentation processes. The best cultivation condition in our study resulted in a squalene productivity of 59.5 tons per year, highlighting its suitability for industrial-scale bioproduction. This selection ensures that the economic feasibility analysis reflects realistic and achievable yields and provides an accurate cost-benefit assessment. In addition, Aurantiochytrium sp. offers advantages such as adaptability to fermentation processes, genetic and metabolic potential, and sustainability by reducing dependence on marine-derived resources. Our updated NPV analysis, based on the Russo et al. techno-economic valuation framework, shows that microbial squalene production is highly viable, yielding an NPV of USD 36.2 million over a 10-year period. This valuation takes into account an initial investment of USD 500,000, a discount rate of 8% and annual operating costs of USD 480,288 covering raw materials, energy, labor and maintenance. Projected annual revenues are estimated at USD 5.47 million, based on a market price of USD 100/kg (see SM). These results confirm the economic competitiveness of microbial squalene production compared to shark-derived sour.
Reviewer 3 Report
Comments and Suggestions for Authors
My opinion about this article not changed, even after long explanations by authors. Actually this MS is about selection optimal conditions for squalene production. Role sharks in this item is minor. I think that sufficient to mention them as a source of squalene in Introduction and Discussion, but not in Results.
There are special IUPAC rules for nomenclature of organic compounds. For squalene (CAS No 111-02-4) preferable IUPAC name is "(6E,10E,14E,18E,22E)-2,6,10,15,19,23-Hexamethyltetracosa-2,6,10,14,18,22-hexaene." More simple variant is 2,6,10,15,19,23-Hexamethyltetracosa-2,6,10,14,18,22-hexaene (without displaying cis/trans configuration of double bonds). In line 33 shown impossible variant, because C6 can’t have two double bonds!
Author Response
My opinion about this article not changed, even after long explanations by authors. Actually this MS is about selection optimal conditions for squalene production. Role sharks in this item is minor. I think that sufficient to mention them as a source of squalene in Introduction and Discussion, but not in Results.
We sincerely appreciate the reviewer’s feedback and respect their perspective. However, we would like to highlight that the discussion regarding sharks is an integral part of the initial proposal of our research. Removing this aspect would significantly impact the overall coherence of the study. Additionally, we must consider that another reviewer specifically requested an improvement in the sensitivity analysis, which further reinforces the need to maintain this discussion.
Moreover, the manuscript has already been accepted by another reviewer in its current format. Making such a substantial modification at this stage could lead to inconsistencies with the feedback provided by the other reviewers, potentially compromising the manuscript’s acceptance. We kindly ask for the reviewer's understanding in this regard.
There are special IUPAC rules for nomenclature of organic compounds. For squalene (CAS No 111-02-4) preferable IUPAC name is "(6E,10E,14E,18E,22E)-2,6,10,15,19,23-Hexamethyltetracosa-2,6,10,14,18,22-hexaene." More simple variant is 2,6,10,15,19,23-Hexamethyltetracosa-2,6,10,14,18,22-hexaene (without displaying cis/trans configuration of double bonds). In line 33 shown impossible variant, because C6 can’t have two double bonds!
Thank you for your valuable feedback regarding the nomenclature of squalene. We have carefully reviewed the IUPAC naming conventions and have made the necessary corrections to ensure accuracy. The incorrect variant previously mentioned in line 33 has been revised accordingly, ensuring that the chemical structure is properly represented.
